# Visualizing Cell Death in Live Retina: Using Calpain Activity Detection as a Biomarker for Retinal Degeneration

**DOI:** 10.3390/ijms23073892

**Published:** 2022-03-31

**Authors:** Soumaya Belhadj, Nina Sofia Hermann, Yu Zhu, Gustav Christensen, Torsten Strasser, François Paquet-Durand

**Affiliations:** 1Cell Death Mechanisms Group, Institute for Ophthalmic Research, Eberhard-Karls-Universität Tübingen, 72076 Tübingen, Germany; soumaya.belhadj@uni-tuebingen.de (S.B.); yu.zhu@uni-tuebingen.de (Y.Z.); gustav.christensen@uni-tuebingen.de (G.C.); 2Graduate Training Center of Neuroscience, Eberhard-Karls-Universität Tübingen, 72076 Tübingen, Germany; nina-sofia.hermann@student.uni-tuebingen.de; 3Applied Vision Research Group, Institute for Ophthalmic Research, Eberhard-Karls-Universität Tübingen, 72076 Tübingen, Germany; torsten.strasser@uni-tuebingen.de; 4University Eye Hospital Tübingen, Eberhard-Karls-Universität Tübingen, 72076 Tübingen, Germany

**Keywords:** retina, calpain, cell death, biomarker

## Abstract

Calpains are a family of calcium-activated proteases involved in numerous disorders. Notably, previous studies have shown that calpain activity was substantially increased in various models for inherited retinal degeneration (RD). In the present study, we tested the capacity of the calpain-specific substrate *t*-BOC-Leu-Met-CMAC to detect calpain activity in living retina, in organotypic retinal explant cultures derived from wild-type mice, as well as from *rd1* and *Rho^P23H/+^* RD-mutant mice. Test conditions were refined until the calpain substrate readily detected large numbers of cells in the photoreceptor layer of RD retina but not in wild-type retina. At the same time, the calpain substrate was not obviously toxic to photoreceptor cells. Comparison of calpain activity with immunostaining for activated calpain-2 furthermore suggested that individual calpain isoforms may be active in distinct temporal stages of photoreceptor cell death. Notably, calpain-2 activity may be a relatively short-lived event, occurring only towards the end of the cell-death process. Finally, our results support the development of calpain activity detection as a novel in vivo biomarker for RD suitable for combination with non-invasive imaging techniques.

## 1. Introduction

Inherited retinal degeneration (RD) is a group of genetic diseases affecting the retina characterized by progressive photoreceptor degeneration and leading to vision impairment and, ultimately, blindness. Mutations in at least 280 genes (https://sph.uth.edu/retnet; information retrieved on 20 March 2022) have been associated with RD. The prevalence of monogenic RD is approximately 1 in 3500 individuals [1].

RD is largely untreatable, and the lack of therapy is due to several reasons: the wide genetic heterogeneity of the disease [2], the difficult delivery of therapeutic compounds across the blood–retinal barrier [3], and the lack of biomarkers to accurately detect the progression of the disease early on. Typically, RD shows a slow progression, with the appearance of symptoms often occurring only in an advanced disease state. Detecting and addressing early phases of pathophysiological cascades is therefore of great importance for therapy development and diagnosis. Moreover, early biomarkers would be of great help to characterize patient populations and to assess treatment efficacy [4].

Most of the mutations causing RD affect genes related to the phototransduction cascade and often cause a dysregulation of cGMP. The toxicity of high levels of cGMP for photoreceptors was already established in the 1970s [5]. The two main known targets of cGMP are protein kinase G (PKG) and CNG channels (CNGC). Notably, the activity of the latter leads to an increased intracellular calcium concentration [6], which may subsequently cause an activation of calcium-activated proteases belonging to the calpain family [7]. In this 15-member protease family, calpain-1 and calpain-2, also known as µ-calpain and m-calpain [8], are activated by micromoles and millimoles of calcium, respectively [9]. While calpain-1 activation mediates synaptic plasticity and neuroprotection, calpain-2 limits the extent of plasticity and may promote neurodegeneration [10,11].

Incidentally, many RD animal models, with a variety of different disease-causing mutations, display extensive activation of calpains in degenerating photoreceptors. This includes the *rd1*, *rd2*, and *rd10* mouse models; the *Rho* S334ter, *Rho* P23H, and WBN/Kob rat models; as well as photoreceptor degeneration induced by treatment with NaIO_3_ or N-methyl-nitrosourea (MNU) [12,13,14,15,16,17,18]. To assess photoreceptor calpain activity, a well-established assay is available that uses the calpain substrate *t*-BOC-Leu-Met-CMAC (CMAC) on unfixed, ex vivo tissue [13,19].

In the present study, we tested the capacity of the CMAC substrate to detect calpain activity in living tissue in organotypic retinal explant cultures derived from *rd1*, *Rho^P23H/+^*, and wild-type mice. An evaluation of retinal cytotoxicity using TUNEL assay indicated that the CMAC substrate was not toxic to photoreceptor cells. Moreover, the CMAC substrate was used to further study the role of calpains in the cell-death mechanism leading to retinal degeneration. Overall, the newly established assay procedure is fast and easy to perform and could potentially lend itself to in vivo biomarker development.

## 2. Results

### 2.1. Calpain Activity Detection in Live Retina

Calpain activity has previously been assessed ex vivo on unfixed retinal tissue sections using CMAC [12,13,19]. Here, we tested whether CMAC could also be used to detect calpain activity in living organotypic retinal explant cultures. The CMAC substrate is processed by endogenous glutathione S-transferase, which conjugates glutathione to the coumarin segment of the molecule [20,21] (Figure 1). Then, the cleavage by calpain results in red shifting and enhancement of the fluorescence emission peak of the MAC-SG product. CMAC is hydrophobic and thus potentially suitable for calpain activity detection in vivo, as it is expected to cross the relevant biological membranes, such as the outer blood retinal barrier [22] and cell membrane [21]. After glutathione conjugation, the resulting MAC-SG product is considered to be cell-impermeable [21].

To test whether the CMAC substrate could be used for the detection of calpain activity in living tissue, we used organotypic retinal explant cultures and exposed them to CMAC for 1 h, 3 h, 6 h, and 24 h (Figure 2A–D; cf. Figure 6). Retinal explants were derived from either wild-type mice or from two distinct RD models: *rd1* and *Rho^P23H/+^* mice.

In *rd1* mouse retinal explants cultivated from postnatal day (P) 5 to P12, an increasing number of photoreceptors (i.e., cells in the outer nuclear layer, ONL) displaying calpain activity was detected when the CMAC incubation time was increased from 1 h to 6 h (Figure 2A–C,E), with the detection rate going up from 0.7% (±0.6, n = 6) to 6.8% (±3.4, n = 9). After 6 h of incubation, the calpain active cell-detection rate appeared to no longer increase. Indeed, after 24 h incubation, the amount of detected positive cells was 4.9% (±2.5, n = 6), which was not significantly different from the 6 h incubation time (Figure 2D,E). The numbers of calpain-positive cells detected in the ONL between the 6 h and 24 h time points corresponded to what was observed in ex vivo retinal sections from the *rd1* animal model at this age [13,17,19]. In the inner nuclear layer (INL), the number of detected calpain-active cells was significantly lower than in the ONL for both the *rd1* and wild-type conditions (Figure 2A–D,F). In wild-type explants, the number of calpain-positive cells remained low throughout the incubation period.

To address the question of whether calpain activity could also be detected in live retina suffering from a dominant mutation in the rhodopsin gene, we further investigated explant cultures derived from heterozygous *Rho^P23H/+^* mice. According to previous studies on this model, rod photoreceptor degeneration starts around P14, peaks at P18–P20, and is almost completed by P31 [23,24]. At P18, calpain activity could also be detected in the *Rho^P23H/+^* mouse model (Appendix A), although the extent of calpain activity was comparatively low, with 1.09% (±0.7, n = 3) positive cells in the ONL. At the same time, the number of dying, TUNEL-positive cells reached 3.8% (±0.7, n = 3) (Appendix A).

### 2.2. Calpain Activity Assay Exhibits Low Toxicity in Live Retina

Because the CMAC substrate had, so far, not been tested in live retinal cultures, the potential toxicity of the compound was investigated. Organotypic retinal explant cultures derived from *rd1* and wild-type mice were treated either with the CMAC substrate or without (2.5% DMSO) at different incubation times: 1 h, 3 h, 6 h, and 24 h. A TUNEL assay [25] was then used for the detection of cell death on fixed tissue sections from these cultures (Figure 3 and Appendix A).

Qualitatively, it was observed that the typical histology of the retina was consistently preserved after all treatments (Figure 2A–D and Figure 3A–D). Additionally, the number of TUNEL-positive cells detected in the ONL was not significantly different between the different incubation periods for *rd1* and wild-type animals (Figure 3E), with approximately 3.1–4.3% detected positive cells in the *rd1* model across incubation times and from 1.0% to 1.6% in the wild-type. Additionally, the number of TUNEL-positive cells detected in the ONL is in accordance with what is usually observed in these models at the same age [12,17,19].

In the INL, the number of TUNEL-positive cells remained negligible in cultures treated with the CMAC substrate from 1 h to 6 h, with approximately 0.5% positive cells detected in the *rd1* model and approximately 0.3% positive cells detected in the wild-type. However, an increase in cell death was observed when cultures derived from both *rd1* and wild-type mice were treated with the CMAC substrate for 24 h, with 1.9% (±0.7, n = 4) positive cells detected in the *rd1* model and 2.0% (±0.1, n = 3) positive cells detected in the wild-type (Figure 3F). This increase in the TUNEL counts was not observed after the same incubation time with the vehicle (DMSO, 2.5%) (Appendix A), suggesting that the observed effect could be due to prolonged exposure to the CMAC substrate.

### 2.3. Calpain Activity and TUNEL Assay Partly Colocalize

To further relate calpain activity with cell death, we studied the colocalization of TUNEL-positive, dying cells with the calpain activity assay. In the ONL, around 3.8% (±1.2, n = 3) of cells were found to be TUNEL-positive (Figure 4), which is in accordance with previous studies [26]. Approximately 6.2% (±0.6, n = 3) of photoreceptors stained positive for the calpain substrate CMAC, which was used as a proxy for overall calpain activity (Figure 4A–C). Quantification of colocalization between TUNEL and calpain activity showed that 1.3% (±0.5, n = 3) of all cells were double-positive for both assays (Figure 4C,D). Hence, only about 20% of the cells exhibiting calpain activity also showed TUNEL positivity. Because the TUNEL assay is thought to label cells at the very end of the cell-death process [27], this implies that around 80% of calpain-positive cells had still not reached the final stages of cell death.

The calpain family comprises 15 isoforms, and the CMAC substrate is not specific to a particular isoform. Because calpain-2 had previously been reported as the isoform driving neurodegeneration [10,11], we performed immunostaining for activated calpain-2 to assess the involvement of this isoform in the neurodegenerative process. While in this situation, 7.2% of the cells in the ONL (±0.1, n = 3) were detected as calpain positive, only 2.7% (±0.2, n = 3) of these cells stained positive for activated calpain-2 (Figure 4E–G). Regarding the colocalization between calpain activity and activated calpain-2, 1% (±0.3, n = 3) of all cells were double-positive (Figure 4G,H). The staining of activated calpain-2 presented as a ring along the periphery of photoreceptor cell bodies, which corroborates the finding that calpains are activated at the cell membrane upon the availability of Ca^2+^ and phospholipids [28] (Figure 4F,J).

A further colocalization experiment was carried out between TUNEL and activated calpain-2. While 3.3% (±1.1, n = 5) of photoreceptors were positive for TUNEL, 2.1% (±0.8, n = 5) of them were positive for activated calpain-2 (Figure 4I–K). Around 2% (±0.7, n = 5) of all photoreceptors were double-positive for TUNEL and activated calpain-2, which accounts for almost all cells positive for activated calpain-2 (Figure 4K,L). This strong overlap between TUNEL and calpain-2 activation indicates that calpain-2 activation occurred only in the final stages of the cell-death process.

### 2.4. Estimating the Duration of Calpain Activity in Individual Dying Photoreceptors

To assess for how long the activity identified with the CMAC substrate (MAC-SG) was detectable in individual ONL cells, we first treated *rd1* retinal explants with CMAC and then reverted to unlabeled culture medium without CMAC for post-labelling incubation periods ranging from 0 to 6 h (Figure 5A–D; cf. Figure 6). Here, the retinal cultures were treated with the CMAC substrate for 6 h, as this was the incubation time for which we observed the strongest signal (cf. Figure 2).

The amount of detected calpain-positive cells decreased sharply with the post-labelling incubation period until it reached a lower threshold at approximately 3 h after labelling. Indeed, at 0 h post-labelling incubation period, 6.8% (±3.4, n = 9) calpain-positive cells were detected, whereas 4.3% (±3.1, n = 6), 2.3% (±1.6, n = 7) and 2.0% (±0.8, n = 7) calpain-positive cells were detected after 1 h, 3 h, and 6 h post-labelling incubation periods, respectively (Figure 5A–E). Based on the logic forwarded in a study by Clarke and colleagues [29], we fitted a one-phase exponential decay model to the calpain activity data. The model estimates a half-life time of a calpain-positive cell of around 0.9 h (best fit prediction) (Figure 5E,F). In other words, in a given dying photoreceptor cell, it will likely take a little less than 1 h for the calpain activity signal to dissipate.

## 3. Discussion

The detection and quantification of cellular calpain activity is highly desirable in the context of many diseases, including neurodegenerative diseases of the retina. In this study, we used the CMAC substrate to monitor calpain activity in living retina and to relate it to cell death and the activity of the calpain-2 isoform. Because the retina is accessible to non-invasive in vivo imaging, using, for instance, scanning laser ophthalmoscopy [30,31], the direct detection of calpain activity appears feasible and could significantly improve our understanding of physiological and pathological processes. Moreover, calpain activity detection could serve as a surrogate biomarker for the study and diagnosis of neurodegenerative diseases.

### 3.1. Detection of Calpain Activity

Numerous assay kits using fluorogenic- or luminogenic-specific calpain substrates are commercially available to detect calpain activity in cell or tissue lysates. However, these assays do not offer cellular resolution of calpain activity [32]. Calpain activity can be detected indirectly by using antibodies recognizing substrate proteins after calpain-specific cleavage. For example, antibodies recognizing the 150 kDa cleaved fragment of α-fodrin have been used to detect calpain-specific fragmentation using the Western blot technique or immunofluorescence [33,34]. Upon Ca^2+^-dependent activation, calpains undergo an autolytic cleavage that reduces their apparent molecular weight by about 2 kDa [28]. This opens another way to assess calpain activity on histological tissue sections via the use of antibodies recognizing only the activated calpain-1 or calpain-2 protease [17]. Moreover, in vivo imaging of calpain activity was enabled in a mouse model that ubiquitously expresses a FRET reporter consisting of eCFP and eYFP and separated by a linker cleavable by calpains [35,36]. As opposed to these approaches, the detection method forwarded in this study opens the possibility to perform in vivo imaging of calpain activity without the use of transgenic animals or antibody detection systems.

The apparent sensitivity of the live tissue assay, as determined by the positive cell detection rate, is similar to that of the established assay using the same substrate on unfixed dead tissue sections [13,19]. Moreover, calpain activity could be detected in organotypic retinal explants derived from two genetically distinct mouse models for retinal degeneration, suggesting a general applicability of the assay in the genetically very heterogeneous RD-type diseases. The mutation underlying the *rd1* mouse phenotype affects the *Pde6b* gene. In humans, PDE6B mutations are responsible for autosomal recessive retinitis pigmentosa (arRP) in about 4–5% of patients [2]. Mutations in the rhodopsin gene are the most common form of autosomal-dominant retinitis pigmentosa (adRP), and the variant RHO^P23H^ is the most common cause of adRP in the United States [37]. It is interesting to note that in the *rd1* model, at P11, i.e., at the onset of degeneration, calpain-active cells outnumbered TUNEL-positive cells by about 2:1. In the *Rho^P23H/+^* model, at P18, a comparable time-point in the progression of retinal degeneration, the calpain-positive cells were outnumbered by TUNEL-positive cells in a ratio of about 1:4. This discrepancy between the recessive *rd1*- and dominant *Rho^P23H/+^* models could point at differences in the execution of cell death, perhaps associated with different kinetics of the various processes that govern the degeneration.

### 3.2. Calpain Activity Assay Exhibits Low Toxicity in Live Retinal Explant Cultures

The fact that the histology of the tissue remained undisturbed after the different incubation times with the CMAC substrate, as well as the TUNEL count in ONL and INL for incubation times up to 6 h, suggests that the CMAC substrate is not toxic to photoreceptors or to the retina in general. This is in line with cell-culture-based studies using the CMAC substrate, which also did not report toxicity of the probe [38,39,40]. The increase in TUNEL counts in the INL after the 24 h incubation time with CMAC in both *rd1* and wild-type mice may suggest that the substrate could be toxic to INL cells after prolonged exposure. Future studies using specific markers for different INL cell types may reveal whether certain cell populations might be sensitive to CMAC.

### 3.3. Calpain-2 Is Activated in the Final Stages of Photoreceptor Cell Death

Several recent studies have shown that individual calpain species may serve distinct roles in vivo [10,41,42]. For instance, calpain isoforms show differences regarding the time window of proteolysis, the pattern of autolysis, and the concentration of Ca^2+^ required for activation [43,44]. Notably, calpain-2 activation requires Ca^2+^ concentrations in the high micro- to millimolar range, compared to calpain-1, which is activated by low-micromolar Ca^2+^ [8]. Calpain activity, as identified by CMAC labeling and calpain-2 immunostaining, has previously been associated with cell death in photoreceptor degeneration [13,17,45]. However, colocalization analysis revealed significant differences between the TUNEL assay, CMAC labeling, and calpain-2. This suggests that the different assays capture dying cells at distinct time-points during the progression of photoreceptor cell death. The TUNEL assay is thought to label cells at the very end of the process of degeneration [27], whereas calpain activity is believed to occur at a slightly earlier stage [6,17]. The low percentage of co-staining between calpain activity and TUNEL observed here suggests that calpain activity occurred distinctly earlier in the timeline of cell death than TUNEL positivity. Interestingly, the overlap between calpain-2 and calpain activity was also relatively low, suggesting that activation of calpain-2 occurred at a later time interval as compared to overall calpain activity. The very high but not complete colocalization between TUNEL and calpain-2 further suggests that calpain-2 activation occurred at a late, nearly final stage of cell death. Together with the fact that calpain-2 activation was restricted to the perinuclear areas of photoreceptors, this raises the question as to how calpain-2 might be activated. A recent study pointed to a potential role of voltage-gated Ca^2+^ channels (VGCC), which are expressed in photoreceptor cell bodies and which might mediate the influx of Ca^2+^ ions required for calpain-2 activation [46]. Combined with the higher availability of Ca^2+^ in the cytosol in advanced stages of cell death caused by the degradation of cell organelles and overall dysregulation of Ca^2+^ homeostasis, this might explain the delayed occurrence of calpain-2 activity.

### 3.4. Calpain Activity: A Short Event in the Photoreceptor Death Process?

The fluorescent signal from the CMAC calpain substrate was extinguished in a time course that appeared to follow an exponential decay. Indeed, previous studies suggest that cell death follows exponential kinetics [29]. The half-life for calpain positivity, estimated to be 0.94 h (best fit prediction), may seem relatively short, given that the total duration of photoreceptor cell death in *rd1* retina was estimated to last around 80 h [27]. Still, because the MAC-SG product is deemed to be cell impermeable [21], it is unlikely that the extinction of the signal was caused by diffusion out of the cell. Instead, the duration of calpain activity is likely a rather short event during the photoreceptor cell-death process.

We observed that even fewer cells were positive for activated calpain-2 than for overall calpain activity. Because CMAC labelling captures the activity of all calpain isoforms present in the retina, including but not exclusive to calpain-2, this discrepancy is to be expected. Furthermore, the calpain activity assay was performed on live retinal explants over a duration of 6 h, whereas calpain-2 staining captures a snapshot of those cells in which calpain-2 was activated precisely at the time-point of fixation. Another reason might be that calpain-2 is inactivated very quickly via autolysis [47,48], which might lead to only small amounts of CMAC being cleaved by calpain-2. Further co-staining experiments, including with other calpain isoforms and other signaling molecules involved in cGMP-dependent cell death, could help to unravel the time course of photoreceptor degeneration more precisely. Due to a lack of available antibodies and the lethality of certain calpain knockout models [49,50], this may require the development of an appropriate methodology.

## 4. Materials and Methods

### 4.1. Animals

C3H HeA *Pde6b^rd1/rd1^* (*rd1*) [51], *Rho ^P23H/+^* [52] and wild-type (WT) mice were used. Animals were housed under standard white cyclic lighting, had free access to food and water, and were used irrespective of gender. Animal protocols compliant with §4 of the German law of animal protection were reviewed and approved by the Tübingen University committee on animal protection (Einrichtung für Tierschutz, Tierärztlichen Dienst und Labortierkunde; Registration No. AK02/19M and AK05/19M).

### 4.2. Organotypic Retinal Explants

The organotypic retinal explant cultures were prepared as described previously [53]. At postnatal (P) day 5, *rd1* animals were killed, and the eyes were rapidly enucleated in an aseptic environment. The entire eyes were incubated in R16 serum-free culture medium (Gibco, Carlsbad, CA) with 0.12% proteinase K (MP Biomedicals, Illkirch-Grafenstaden, France) at 37 °C for 15 min to allow for preparation of retinal cultures with retinal pigment epithelium (RPE) attached. Proteinase K was inactivated with 10% FCS (Gibco) in R16 medium, and thereafter, the eyes were dissected aseptically in a Petri dish containing fresh R16 medium. The anterior segment, lens, vitreous, sclera, and choroid were carefully removed by fine scissors, and the retina was cut perpendicular to its edges, giving a cloverleaf-like shape. Subsequently, the retina was transferred to a culture dish with membrane insert (COSTAR, NY), with the RPE layer facing the membrane. The insert was put into a six-well culture plate and incubated in R16 medium with supplements at 37 °C. The full volume of nutrient medium, 1 mL per dish, was replaced with fresh R16 medium every second day. The procedures for retinal explant cultures derived from *Rho ^P23H/+^* animals were the same as for *rd1* mice, except that the *Rho* mutants were killed at P14 and then cultured for four days until P18.

### 4.3. TUNEL Assay

A terminal deoxynucleotidyl transferase dUTP nick end labeling (TUNEL) assay [25] was performed using a fluorescein kit from Roche Diagnostics (Mannheim, Germany). Sections were incubated in blocking solution (1% BSA, 10% normal goat serum, 1% fish gelatin) for 1 h after 5 min incubation with alcohol acetic acid mixture (62% EtOH, 11% Acetic Acid, 27% H_2_O). Sections were then stained using the kit as per the manufacturer’s instructions.

### 4.4. Immunohistochemistry

The fixed cryosections were dried at 37 °C for 30 min and washed in PBS for 10 min. Then, blocking solution (10% NGS and 1% BSA in 0.3% PBST) was applied to the tissue and incubated for 1 h at room temperature (RT). The anti-calpain-2 primary antibody (Abcam, Cambridge, UK) was diluted 1:300 in blocking solution and incubated with the sections overnight at 4 °C. The sections were then rinsed with PBS three times for 10 min each. The secondary antibody (goat anti-rabbit Alexa Fluor 488 or 568, Molecular Probes, Eugene OR) was diluted 1:500 in PBS and incubated with the sections for 1 h at RT in the dark. The sections were rinsed three times with PBS for 10 min each, mounted, and stored at 4 °C for at least 30 min before imaging.

### 4.5. Detection of Calpain Activity in Retinal Explants

The cultures derived from the *rd1* mouse model were treated for variable incubation periods (1 h, 3 h, 6 h, and 24 h) with the substrate tertiary-butyloxycarbonyl-leucine-methionine-7-amino-4-methylcoumarin (abbreviated to *t*-BOC-Leu-Met-CMAC, or in short: CMAC) (Invitrogen, Carlsbad, CA, USA) at a concentration of 50 μM in the culture medium. CMAC was diluted in the culture medium at P11 for a 24 h incubation or at P12 for the other conditions (1 h, 3 h, and 6 h). At the end of the incubation time, the cultures were stopped by 45 min fixation in 4% PFA. This was followed by graded sucrose cryoprotection, embedding in Tissue-Tek O.C.T. compound (Sakura Finetek Europe, Alphen aan den Rijn, Netherlands)-filled boxes, and collecting of 12-μm-thick retinal cross sections on a Thermo Scientific NX50 cryotome (Figure 6).

In a second set of experiments, the period of time that a given cell would remain positive for calpain activity was investigated. To this end, retinal cultures were incubated at P12 for 6 h with the CMAC substrate (Invitrogen, Carlsbad, CA, USA) at a concentration of 50 μM in the culture medium. Thereafter, the medium containing the CMAC substrate was replaced by fresh culture medium for different incubation times: 1 h, 3 h, and 6 h. Afterwards, the cultures were stopped by 45 min fixation in 4% PFA and further processing as above. The cultures derived from the *Rho^P23H/+^* mouse model were treated with the CMAC substrate at P18 for 6 h at a concentration of 50 µM in the culture medium. Afterwards, the cultures were stopped by 45 min fixation in 4% PFA and further processing as above (Figure 6).

**Figure 6 ijms-23-03892-f006:**
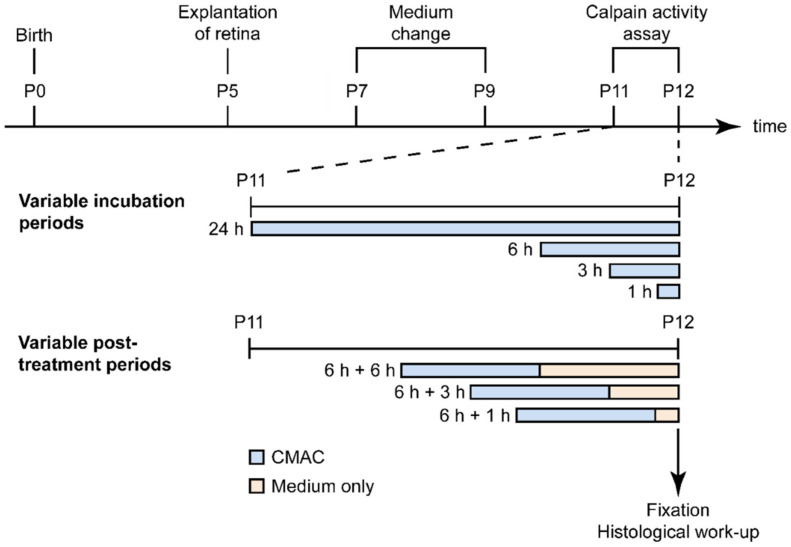
**Overview of experimental paradigms for calpain activity detection in retinal explants.** The organotypic retinal explant cultures were derived from either wild-type or *rd1* animals at postnatal day (P) 5. The culture medium was changed every second day, and retinal explants were cultivated until P12. In the first set of experiments, the CMAC substrate was diluted in the culture medium between P11 and P12 and incubated for variable periods, ranging from 1 h to 24 h. In the second experimental paradigm, the retinal explants were incubated with CMAC substrate for 6 h; then, the medium was replaced by fresh culture medium without CMAC substrate, and the incubation was continued for another 1 h, 3 h, or 6 h. At P12, the cultures were ended by fixation in 4% PFA.

### 4.6. Microscopy, Cell Counting, and Statistical Analysis

Light and fluorescence microscopy were performed at RT on an Axio Imager Z.1 ApoTome microscope equipped with a Zeiss Axiocam MRm digital camera (Zeiss, Oberkochen, Germany). Images were captured using Zeiss Zen software; representative pictures were taken from central areas of the retina using a 20x/0.8 Zeiss Plan-APOCHROMAT objective. For quantifications, pictures were captured on three entire sagittal sections from at least three different animals. The average area occupied by a photoreceptor cell (i.e., cell size) was determined by counting 4′,6-diamidino-2-phenylindole (DAPI)-stained nuclei in nine different areas of the retina. The total number of photoreceptor cells was estimated by dividing the ONL area by this average cell size. The number of positively labelled cells in the ONL was counted manually. Errors in graphs and text are given as standard deviation (STD).

The data on calpain-positive cells in the ONL after variable post-treatment periods were used to fit a one-phase exponential decay model (N(t) = N_0_e^−λt^) using GraphPad Prism (GraphPad software, San Diego, CA, USA). This resulted in a baseline value, N_0_, of 6.821 and a decay rate, λ, of 0.7362 (R^2^ = 0.4150). A half-life time for calpain positivity of 0.9415 h (best fit prediction) was calculated as ln(2)/λ (see model in red; Figure 6).

## 5. Conclusions

The results presented in this study support a further development of the CMAC substrate as a biomarker for cell-death detection. Although pharmacodynamics, pharmacokinetics, and toxicity of this calpain substrate need to be further investigated, in the retina, this method could potentially enable direct, non-invasive detection of calpain activity in vivo with single-cell resolution using techniques such as adaptive optics scanning laser ophthalmoscopy (AO-SLO) [54]. Here, an infrared two-photon excitation system could be used to avoid UV-light stimulation. Alternatively, a calpain-specific peptidic substrate could be combined with a near infrared fluorophore to enable single-photon stimulation. Taken together, our results suggest that the role of calpain in the process of photoreceptor cell death may be more complex than previously assumed, with individual isoforms being active in slightly distinct temporal stages of cell death. Other than biomarker applications, understanding the temporal progression of calpain activities during photoreceptor cell death may also have direct implications in the development of therapeutic approaches for RD-type diseases.

## Figures and Tables

**Figure 1 ijms-23-03892-f001:**
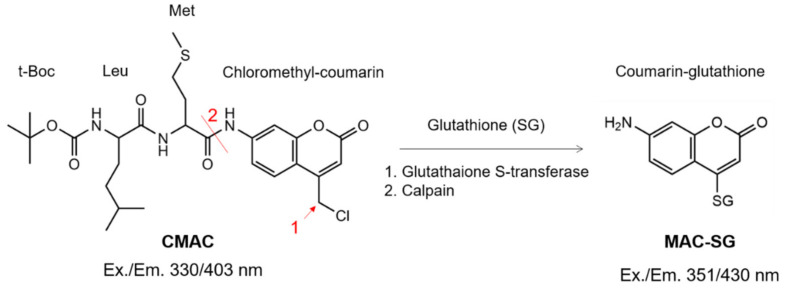
**Molecular structure of the calpain substrate CMAC and product MAC-SG.** The calpain activity assay with the CMAC substrate involves an initial glutathione conjugation to the coumarin segment at the position indicated by arrow 1. Afterwards, calpain cleaves the peptide bond at position 2, unquenching the coumarin-glutathione product and red shifting the excitation and emission peaks.

**Figure 2 ijms-23-03892-f002:**
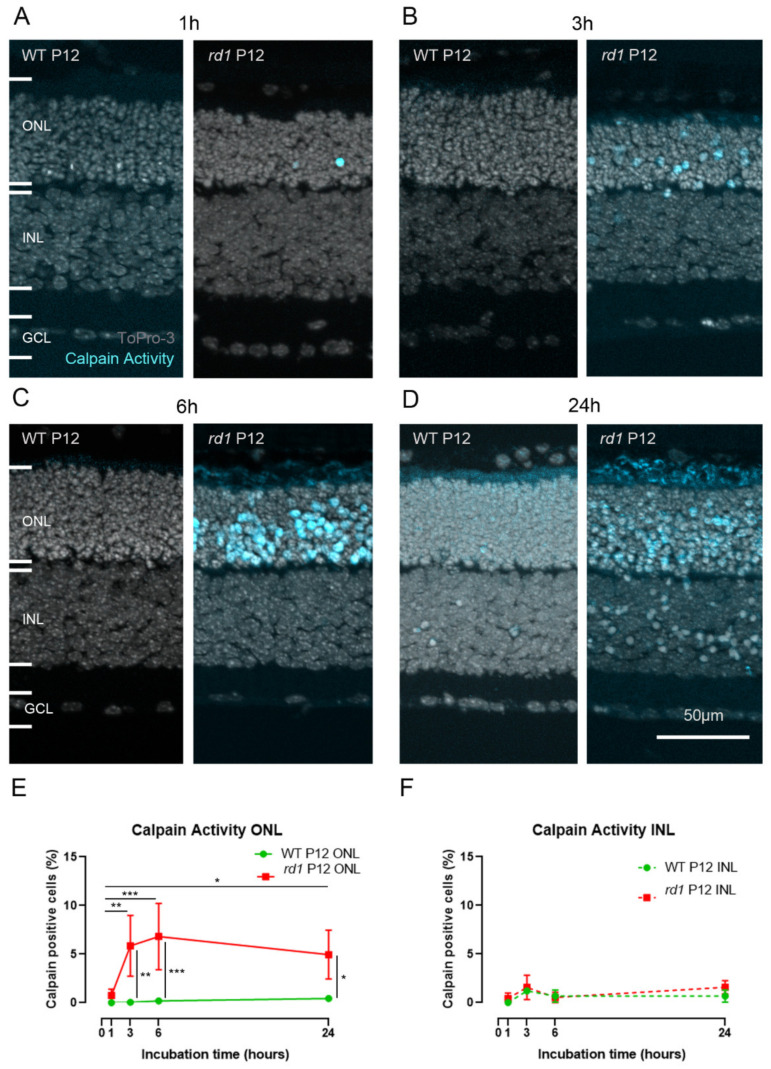
**Live tissue detection of calpain activity in the *rd1* mouse model.** Organotypic retinal explant cultures were incubated with 50 µM CMAC for 1 h (**A**), 3 h (**B**), 6 h (**C**), or 24 h (**D**) before the culture was ended by fixation. To-Pro-3 was used as nuclear counterstain. The number of calpain-positive cells detected in the *rd1* outer nuclear layer (ONL) increased from 1 h to 6 h to reach a plateau by 24 h (**E**), whereas in the inner nuclear layer (INL), no specific trend was observed (**F**). Images representative of results obtained from six to nine independent retinal explant cultures; error bars indicate STD; statistical analysis: two-way ANOVA with Holm–Sidak’s multiple comparisons test; * = *p* < 0.05; ** = *p* < 0.01; *** = *p* < 0.001; GCL = ganglion cell layer.

**Figure 3 ijms-23-03892-f003:**
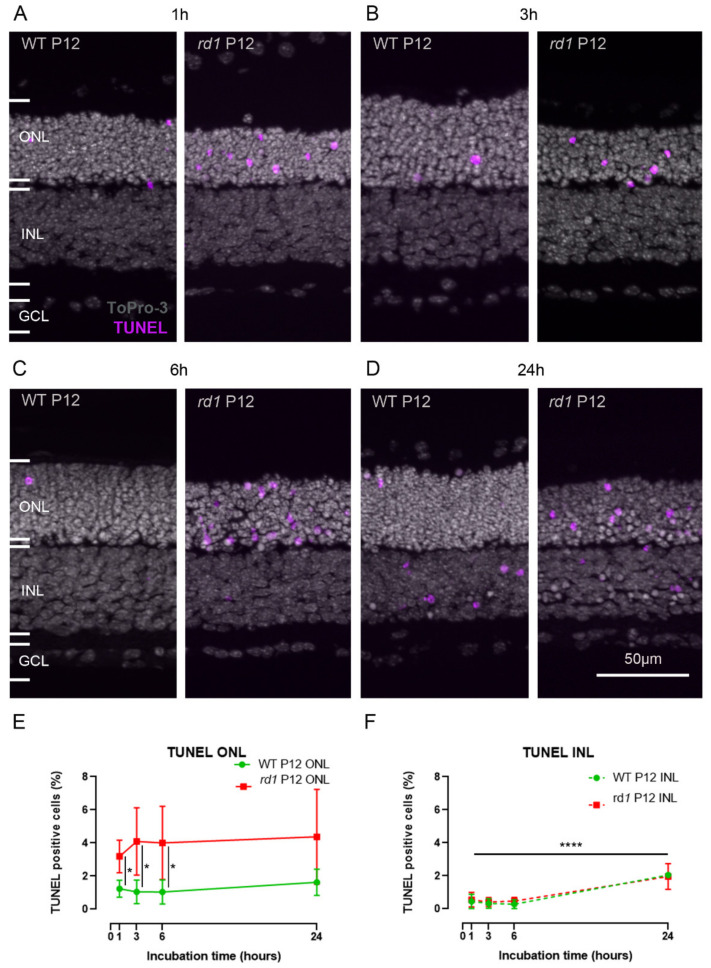
**Cell death in live tissue treated with the CMAC substrate.** Organotypic retinal explant cultures derived from the *rd1* mouse model were incubated with 50 µM CMAC for 1 h (**A**), 3 h (**B**), 6 h (**C**), or 24 h (**D**). A TUNEL assay for cell death detection was carried out on fixed tissue sections. Percentage of TUNEL-positive cells in the outer nuclear layer (ONL) (**E**) and the inner nuclear layer (INL) (**F**). To-Pro-3 was used as nuclear counterstaining. Images are representative of results obtained from four to nine independent retinal explant cultures; error bars indicate STD; statistical analysis: two-way ANOVA with Holm–Sidak’s multiple comparisons test; * = *p* < 0.05; **** = *p* < 0.0001; GCL = ganglion cell layer.

**Figure 4 ijms-23-03892-f004:**
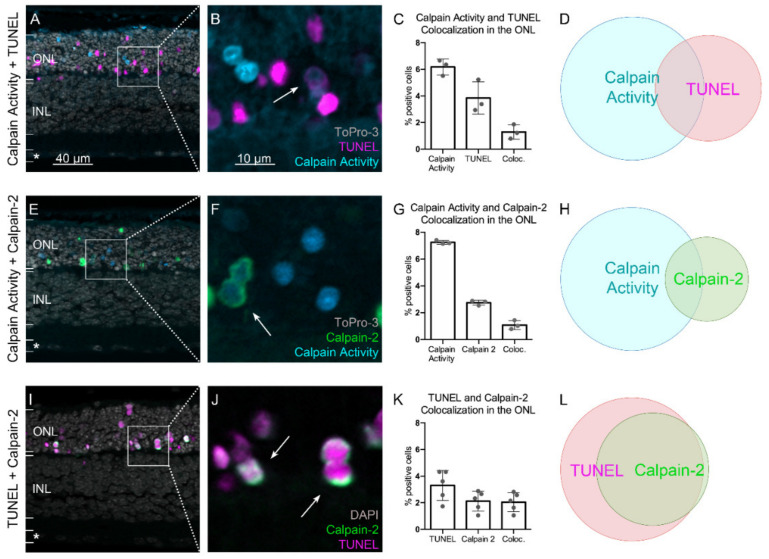
**Colocalization of the TUNEL assay, calpain activity assay, and calpain-2 immunostaining.** (**A**–**D**) Comparison between calpain and calpain-2 activation. (**E**–**H**) Comparison between calpain activity and the TUNEL assay. (**I**–**L**) Comparison between calpain-2 activation and the TUNEL assay. The arrows highlight cells showing colocalization. (**D**,**H**,**L**) Venn diagrams visualizing the relative proportion of cells positive for the respective staining, as well as the overlap between different stainings (colocalization). Images representative of results obtained from three to five independent retinal explant cultures derived from the *rd1* mouse model. Error bars indicate STD. ONL = outer nuclear layer, INL = inner nuclear layer, asterisk = ganglion cell layer.

**Figure 5 ijms-23-03892-f005:**
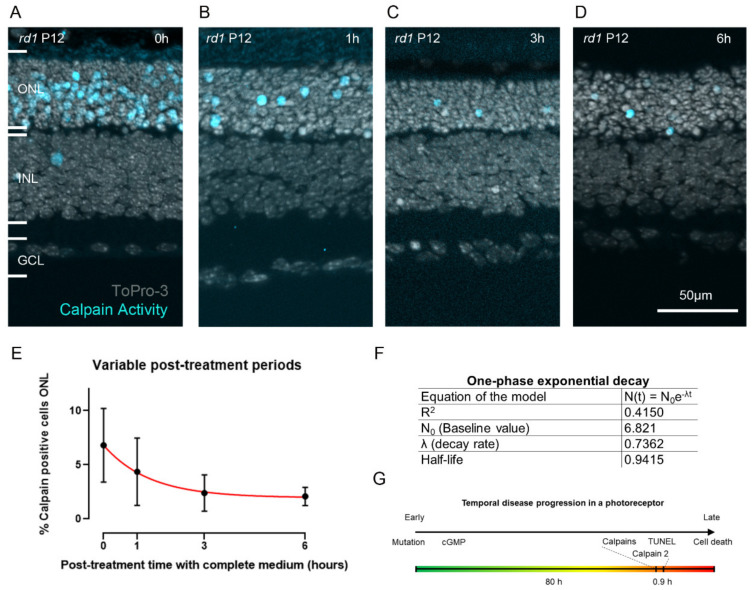
**Estimation of the duration of calpain activity during photoreceptor cell death.** (**A**–**D**) Organotypic retinal explant cultures derived from the *rd1* mouse model were treated with the CMAC substrate for 6 h. The medium containing CMAC was then replaced by plain, unlabeled medium for varying durations: 0 h (**A**), 1 h (**B**), 3 h (**C**), or 6 h (**D**). (**E**) Quantification of calpain-positive cells in the outer nuclear layer (ONL). The number of detected calpain-positive cells over time followed an exponential decay. Red line shows model fit. (**F**) Table showing model equation and parameters. (**G**) Putative timeline for disease progression in an individual photoreceptor cell. Error bars indicate STD. INL = inner nuclear layer, GCL = ganglion cell layer. Images representative of results obtained from six to nine independent retinal explant cultures per condition.

## Data Availability

The raw data supporting the conclusions of this article will be made available upon request.

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
