# Peer review of "Visualizing Cell Death in Live Retina: Using Calpain Activity Detection as a Biomarker for Retinal Degeneration"

_ijms, 2022, doi:10.3390/ijms23073892_

Round 1
Reviewer 1 Report
The article entitled” Vizualizing cell death in live retina: Using calpein activity detection as a biomarker for retinal degeneration” by Soumaya Belhadj et al reported the interest of using calpein as a biomarker for retinal degeneration.
The scientific value is real but unfortunately the authors did only three or four experiment to draw a conclusion. Incorrectly they made statistical calculation with n=3 or n=4 which is far from the minimal number to use the simplest calculation. In figure 2E (increasing number of receptors) the mean and the SD have been calculated with 4 values (mean=7.4% ± 5.1), the SD is far above what could be observed with reproducible experiments. Moreover, it would be better to express results in absolute numbers rather than in percentages.
The authors can resubmit an article with a reasonable number of results at least six for each experiment.
Reviewer 2 Report
Well written manuscript in which authors describe their experiments to measure CMAC, calpain-2 and TUNEL assay in living organotypic retinal explant cultures. Figures are clear and good to interpret. I only have some remarks/ suggestions:
Figure 2. There is a lot of variation in ONL calpain (rd1 P12 ONL) activity compared to WT P12 ONL, which has no apparent variability. Is there simple no calpain activity measured / present in ONL, are error bars missing? How are negative results interpreted (i.e. the discrimination between a negative result or a low quality experiment)
I am a bit los in chapter 3.3. Please elaborate a bit more on the proposed mechanism in which calpain-2 activity precedes in cells with positive calpain-2 immunostaining / TUNEL assay. Line 284 / figure 5 mentions a 6 hour measurement period for signals to fade. TUNEL assay is measured over a 24h period and calpain-2 immunostaining is a snapshot but has a very large overlap with the TUNEL assay (figure 4) Authors suggest a temporal mechanism and suggest a role for the difference in calpain-2 staining and calpain activity for the different isoforms. Are all 15 calpain isoforms expressed in the retina or only some of them? If more isoforms are expressed, these could be expressed in cells that do not have calpain-2 expression or expressed at a different timepoint. Is there any evidence for that experimentally or from literature? Additionally, how certain is it that the calpain activity positive cells actually die given the limited overlap with both TUNEL assay and calpain-2 immunostaining?
Round 2
Reviewer 1 Report
The authors increased the number of experiments and answered all questions